# Public Health Management of the COVID-19 Pandemic in Australia: The Role of the Morrison Government

**DOI:** 10.3390/ijerph191610400

**Published:** 2022-08-20

**Authors:** Stephen Duckett

**Affiliations:** Melbourne School of Population and Global Health, University of Melbourne, Melbourne 3010, Australia; stephen.duckett@unimelb.edu.au

**Keywords:** Australia, public administration, public health policy, vaccination, COVID

## Abstract

The Australian Commonwealth government has four health-related responsibilities during the SARS-CoV-2 pandemic: to provide national leadership; to manage external borders; to protect residents of residential aged care facilities; and to approve, procure and roll-out tests and vaccines. State governments are responsible for determining what public health measures are appropriate and implementing them—including managing the border quarantine arrangements and the testing, tracing, and isolation regime—and managing the hospital response. This paper analyses the national government’s response to the pandemic and discusses why it has attracted a thesaurus of negative adjectives.

## 1. Introduction

Australia’s management of the COVID-19 pandemic has attracted praise, but external observers often erroneously attribute this to the actions of the Commonwealth (national) government alone. Australia is a federation and so all governments had a role in fighting the pandemic.

The re-election of the Liberal-National Coalition government, led by Scott Morrison, was a surprise outcome of Australia’s 18 May 2019 election [1]. For most of its term Australian politics was dominated by the response to the SARS-CoV-2 virus which causes COVID-19. This paper discusses how the Morrison government managed its health sector-related COVID-19 responsibilities. In brief, the answer is poorly. It uses documentary analysis to identify key decisions of policy makers and contemporary accounts of policy processes to assess the effectiveness of the management of the first two and a half years of the pandemic.

This paper presents the first comprehensive review of the management of the pandemic at the national level in Australia. It identifies the manifold shortcomings and suggests directions for addressing those weaknesses. It focuses on the public health response: the Morrison government’s economic response to the pandemic, including its JobKeeper program designed to protect incomes and mitigate the pandemic’s impact on some industries, is discussed elsewhere [2].

Overall, the outcomes of COVID-19 in Australia have been good, with just over 5 million cases and about 8000 deaths at the time of writing (June 2022), with deaths in the low range internationally (see Table 1). However, multifactor standardisation reveals Australia’s overall comparative performance against international peers was not so strong [3,4]. Most deaths occurred in 2022 following the progressive lifting of restrictions after the release of the National Plan.

The comparatively low number of deaths is a result of public health measures—especially ‘lockdowns’ and movement restrictions implemented differentially across the country [5]—which suppressed infection rates in the first two years of the pandemic when the COVID-19 strains were more virulent. About 18,000 deaths were averted by the actions of state and federal governments [6], supported by the Australian public, which endured lockdowns and border controls.

Importantly, the impact of the pandemic was experienced unevenly, both economically and in terms of health effects, with poorer people—particularly women—faring worse [7,8,9]. There were over 140 deaths per 100,000 population in the poorest socio-economic quintile compared to about 40 in the wealthiest quintile. On an age-standardised basis there was 2.6 times the number of deaths in poorer areas compared to wealthier ones [10]. Infections in the 2020 waves in Victoria were substantially higher in more disadvantaged postcodes [11].

The vaccine rollout was an area of clear Commonwealth Government responsibility. It was described as a ‘phenomenal failure in public administration’ [12] and ‘the worst national public policy failure in modern Australian history’ [13]. The ‘incompetent management of the federal government’s responsibilities during the pandemic’ was identified by Labor’s National Secretary as the second of eight reasons why the Coalition lost the 2022 election [14]. Widely publicised delays in the rollout produced Australia’s 2021 word of the year: ‘strollout’ [15]. The Senate Select Committee on COVID-19, where the Government was in a minority, concluded:

To date, the Australian Government’s response has been characterised by poor preparation, a refusal to take responsibility and provide national leadership—including in areas of clear Commonwealth constitutional responsibility—a failure to learn lessons as the pandemic progressed, and significant failures of implementation with, at times, catastrophic consequences [16]

States did not always manage their responsibilities well either [17]. There were quarantine breaches [18], most notably, and very publicly, in Victoria [19,20], and failures in managing cruise ship arrivals [21], in engaging culturally and linguistically diverse communities [22], and in testing and tracing [23]. Despite this, state-directed public health measures—including movement restrictions—meant that infection numbers and rates were low by world standards [3], as were deaths [6]. For most of 2020 and 2021, it was state implementation of public health measures that led to Australia’s good overall outcomes in the pandemic, not Commonwealth Government actions. The Morrison government can claim little credit for Australia’s success. In fact, it mostly hindered the states’ successful responses to the pandemic or bungled its own.

## 2. A New Virus—Four Roles for the National Government

Australia is a federation of six states and two territories, with state and territory governments and the national (‘Commonwealth’) governments all having a role to play in health care. There is no neat dividing line between Commonwealth and state roles—indeed Australian federalism has been described as like a ‘marble cake’ [24].

The Constitution itself lists some Commonwealth areas, that is domains where, if the Commonwealth Parliament passes a law, it will override state laws. The most relevant area is quarantine, assigned to the Commonwealth at Federation [25]. The constitution also gives the Commonwealth power over ‘hospital benefits’, although public hospitals (which provide most hospital care) are creatures of the states. The Commonwealth has acquired income and sales tax powers, creating a ‘vertical fiscal imbalance’ where the Commonwealth raises more money than it requires for the services it provides and the states are in the reverse situation. The Commonwealth contributes around 45 per cent of the costs of running state public hospitals but has effectively no say in their management. The Commonwealth is also responsible for aged care. Areas of policy not explicitly assigned to the Commonwealth in the constitution are the responsibility of states; public health is in that category.

The Australian constitution and the traditional division of Commonwealth and state roles provide a framework for identifying the roles of the two levels of government during the pandemic. The Commonwealth Government has four health-related responsibilities during the SARS-CoV-2 pandemic: to steer the response through effective national political leadership; to manage external borders; to protect residents of residential aged care facilities; and to approve, procure and distribute personal protective equipment, tests and vaccines, and to ensure equity of the vaccination rollout. State governments are responsible for determining what public health measures are appropriate and implementing them—including managing the border quarantine arrangements and the testing, tracing, and isolation regime—and managing the hospital response. States are also responsible for developing strategies to mitigate potential mental health and other effects of restrictions, and to address equity issues in the management of health outcomes and care [26,27,28,29,30,31]. The regulation of workplace health and safety was principally covered by state public health orders.

## 3. Effective National Leadership

The SARS-CoV-2 virus (‘coronavirus’) was first identified in Wuhan in December 2019; Australia had its first case in January 2020. Figure 1 shows the Australian pandemic timeline.

An effective response to the SARS-CoV-2 pandemic required a range of strategies and skills. Sagan, et al. [32] identify ‘steering the response through effective political leadership’ as the first of their list of twenty strategies. Sriharan, et al. [33] have identified three ‘crisis leadership’ competencies required for effective leadership during pandemics: task competencies (e.g., communication and collaboration); adaptive competencies (e.g., decision-making); and people competencies (e.g., empathy and awareness). Unfortunately, all three competencies were weak or missing at the national level, which resulted in ineffective decision-making (a failed National Cabinet), the undermining of state strategies, and a political—rather than evidence-based—National Plan, which the Government was unwilling to revise as circumstances changed.

### 3.1. The National Cabinet

Effective government in a federation requires instruments of collaboration and coordination. The First Ministers (the Prime Minister, state premiers, and territory chief ministers,) and all government portfolios have traditionally had formal mechanisms for Commonwealth–state discussion and decision-making, including meetings of ministers (ministerial councils) and meetings of the chief executive officers of the relevant portfolios. Subcommittees of the bureaucratic bodies also exist for defined areas, so there is a formal coordination mechanism in public health called the Australian Health Protection Principal Committee which brings together all state and territory chief public health officers and the Commonwealth equivalent (the Chief Medical Officer).

Even before the pandemic, there had been criticism of the operation of these coordination mechanisms as being slow, cumbersome and bureaucratic [34].

Given that pandemics do not respect borders, a key ‘task competency’ in a pandemic is collaboration [33]. This is especially true in the Australian federation, since ‘the public health crisis could not be effectively met without drawing on the powers, knowledge and capacities of both levels of government’ [35].

Early March 2020 saw increasing event cancellations [36], and states began to make unilateral calls about the pandemic response [37,38,39,40]. Media attention focused on state premiers and chief health officers as the key decision makers and sources of information about the pandemic response, with the Prime Minister and the Commonwealth Health Minister, Greg Hunt—a long-standing politician and former McKinsey consultant, with family links to the health system (Knott 2017)—relegated to a catch-up role. The Prime Minister responded with a typical ‘centralisation’ and credit-claiming strategy [41], creating a National Cabinet on 13 March 2020, and, in a secret and unprecedented move, the next day he was also sworn in as a Health Minister able to exercise all of the statutory powers assigned to that role, while Greg Hunt continued as the sole public face of the portfolio.

The National Cabinet dealt the Prime Minister into discussions of state decisions and gave the states political cover for difficult choices in the early stages of the pandemic. However, because it was set up in haste, there were no real rules for its operation. It had no power to implement decisions—that power still rested with each of the participants; there was no collective accountability to the public through any of the parliaments; and its workings were not transparent [42]. Initially, the National Cabinet did lead to coordination and consistency of decision making. New South Wales and Victoria, for example, were lock step in introducing strict public health measures such as movement restrictions (‘lockdowns’) in response to the first wave, a coherence strengthened by a decision of the National Cabinet. This unity was facilitated as there was agreement that, in the absence of any vaccine, fighting the virus was the only effective health response and economic considerations were not yet seen by any party as in conflict with health considerations.

However, when, toward the end of the first wave, state policies began to diverge from the national consensus, any outcome of a National Cabinet meeting was a ‘decision’ in name only. Behind the fig leaf of unity, each state and territory went its own way, while the Commonwealth ran a critique from the sidelines.

Hyped as ‘one of the most amazing achievements of the Federation in Australia’s first 200 years’ by Health Minister Greg Hunt [43]—with some academic commentators also lauding its virtues [35,42,44]—the ‘cabinet’ was not a cabinet at all, at least as far as the Administrative Appeals Tribunal was concerned. Crucially, the Tribunal noted

In particular, it is evident that members of the National Cabinet did not regard themselves as bound to support decisions made at the National Cabinet irrespective of their own views, and that at times they acted in ways that were inconsistent with the National Cabinet decisions (Patrick and Secretary, Department of Prime Minister and Cabinet (Freedom of Information) (2021) AATA 2719 at paragraph 190)

Not only did members of the National Cabinet not feel bound by cabinet solidarity—a critical element of a properly functioning ‘Cabinet’ [45]—Commonwealth ministers actively undermined decisions taken by the Prime Minister’s National Cabinet colleagues. This white-anting, which started almost immediately and continued throughout the pandemic, delegitimised public health measures, especially movement restrictions such as lockdowns and state border closures. Canberra-based journalists were given privileged ‘off-the-record’ briefings as part of the Prime Minister’s ‘sole aim of controlling and owning the daily narrative to the single advantage of the most political and ambitiously driven prime minister the country’s ever seen’ [46].

In 2020 the undermining of state action included then Commonwealth Education Minister Tehan attempting to induce private schools to open, despite state closure orders [47]; the Prime Minister placing overt pressure on states to end lockdowns early [48,49]; and the Government intervening to support mining oligarch Clive Palmer’s bid to overturn Western Australia’s border closures [50,51,52]. This action featured prominently in Western Australia during the federal election and undoubtedly contributed to Labor’s success in that state.

The following year, the Prime Minister continued his jawboning to argue states should open up quickly after the release of the 2021 National Plan [53]. When Victoria again pursued a COVID-zero strategy in 2021, Commonwealth Treasurer Frydenberg refused to renew Victorian access to the business support scheme, JobKeeper [54], adding financial pressure to the Government’s campaign to reduce state-imposed public health measures. Treasurer Frydenberg was labelled ‘the Treasurer for New South Wales’, during his campaign for re-election [55], an epithet that may have contributed to his defeat.

Pressure on the states to relax their public health measures continued into 2022 [56,57]. The Prime Minister also undermined the social license for a strong state public health response by ‘dog-whistling’ support for anti-vax demonstrators [58,59]. Coalition back-benchers peddled dangerous alternative ‘treatments’, attacked mask mandates, and espoused anti-vaccination sentiments, with only weak rebukes from the Prime Minister [60,61,62,63], further undermining state public health messaging.

Dominating the Prime Minister’s rhetoric in 2020 [64], and underlying the Commonwealth’s attacks on the legitimacy of state public health measures, was a predisposition to emphasise the impact of restrictions on business, and to frame the pandemic response as a trade-off between public health and the economy, and the general tendency of conservative governments and people of a conservative inclination to be suspicious of any restrictions on personal freedoms [65]. The research evidence for an economy-health trade-off is weak, and still influenced by short-run effects [66,67] and the methodological difficulty of disentangling self-imposed restrictions and those imposed by government. [68,69]. The Government’s ‘health versus economy’ narrative flew in the face of advice from Treasury modelling which concluded, in the context of an unvaccinated community, that:

Continuing to minimise the number of COVID-19 cases, by taking early and strong action in response to outbreaks of the Delta variant, is consistently more cost effective than allowing higher levels of community transmission, which ultimately requires longer and more costly lockdowns [70]

Independent academic modelling also showed the benefit of quicker, harder lockdowns, on both health and economic grounds [71,72,73]. However, the trade-off debate was not unique to Australia. Early in the pandemic, when the nature of the interaction of economic and public health effects was still being debated and public perceptions of preferences still being explored [74], fear of economic effects influenced policy choices in many countries [75].

A policy which particularly attracted the Morrison government’s ire was state border closures, especially those imposed by states with Labor governments. Border closures were announced early in the pandemic. Tasmania, with a Liberal government, closed its border on 19 March 2020; within a week the Northern Territory, Queensland, South Australia, and Western Australia had followed suit. Both Victoria and New South Wales eventually used border closures as a public health control, and in Victoria an internal border between Melbourne and the rest of the state was also imposed [76]. Border closures were obviously disruptive to border communities and to those who wanted to travel for business and family reasons [77,78]. Clive Palmer tried to overturn the Western Australia’s border closure, but the High Court ruled it a legitimate use of the state’s public health power [79].

Importantly, framing of policy choices is likely to affect public perceptions [80], and so the continued public attacks by Commonwealth ministers, including by the Prime Minister, probably contributed to undermining the perceived legitimacy of state-imposed public health measures. Even though state public health measures were supported by the public [81], especially in Western Australia where the state government was overwhelmingly returned in the 2021 state election [82], Commonwealth government criticism weakened the social mandate for state public health measures. As a consequence, restrictions were lifted earlier than the public health evidence suggested they should have been, resulting in increasing infections and deaths, especially after the release of the National Plan in August 2021.

The Morrison government’s prioritisation of perceived economic impacts over public health considerations—including hospital capacity—may reflect the relative responsibilities of the Commonwealth and state governments: ‘where you stand depends on where you sit’ [83]. However, the Government’s opposition to lockdowns and mask mandates was probably also driven by an ideological predilection for less government intervention—rather than public health evidence [84]—and the pressure it was under from business lobbyists opposed to lockdowns and continuing constraints on their activities [85]. The attempt to shift blame about the impact of lockdowns on the economy was an example of ineffective blame behaviour seen in other federations during the pandemic [86]. A key element of the Morrison government’s economic response was the poorly-designed JobKeeper program [87,88,89], which excluded key industries affected by the pandemic, contributing to the inequitable impact of COVID-19.

The creation of a National Cabinet, or like body, was an essential prerequisite to an effective response to the pandemic: the powers of the Commonwealth and the states needed to be ‘bundled together’ [90] in a mutually reinforcing way. The National Cabinet could have been a forum for an effective dialogue between leaders. Perhaps it could have functioned to promote an effective dialogue with the public about the nature of the SARS-CoV-2 risk [91]. However, although the meetings took place, there was no consensus-building dialogue and no comprehensive or coherent national response to the crisis. Instead, the Commowealth and the states pursued inconsistent policies which hindered the management of the pandemic.

The Morrison government’s undermining and criticism of state responses did not resonate well with voters in key states. In fact, Labor’s National Secretary attributed ‘Cabinet-wide partisan attacks on state and territory governments throughout COVID-19 which particularly alienated voters in Victoria and Western Australia as the third of eight reasons why the Coalition lost the election [14].

### 3.2. The National Plan

The Prime Minister obtained National Cabinet endorsement for the ‘National Plan to transition Australia’s National COVID-19 Response’ on 6 August 2021 [92]. The plan was developed in the absence of consistent national information about the pandemic, a lacuna which had still not been addressed in the third year of the pandemic [93].

The plan was obscured by a ‘veil of vagueness’ [94], and was hedged in qualifiers. It only applied in a given state when Australia as a whole—and that state—met specific targets, a constraint that was subsequently ignored in Morrison government rhetoric. There was an approximation tilde in front of the first vaccination target, perhaps to disguise the fact that the target referred to the moment when people were vaccinated, not when the immune response had kicked in—a significant difference when vaccinations were increasing rapidly—even though the modelling had been based on the point at which people were effectively immunised, that is, approximately two weeks after jabs went into arms [95]. The list of measures for each phase was preceded by the uncertainty verb ‘may’, and the measures themselves expressed in vague terms (‘Lockdowns less likely but possible’, ‘Ease restrictions on vaccinated residents (TBD);’). Key bones of contention, such as state border closures, were not addressed in the plan. Nor was there any mention of a ‘recovery’ phase, where evaluation occurs and structures established to ensure lessons are learned [96,97].

The plan foreshadowed a progressive relaxation of restrictions once two vaccination thresholds for the population 16 years and older were attained: approximately 70 percent and at 80 percent (peculiarly expressed as ‘greater than or equal to’ 80 per cent), representing about 56 percent and 64 percent of the total population respectively. These low population thresholds were well below those thought to be required to achieve herd immunity [98], and were presumably politically determined, with the aim of bringing an end to state public health measures as soon as possible. The first limb of the first threshold (70% of the 16 and over population vaccinated) was reached on 20 October 2021, but New South Wales opened up before that date, ignoring the National Plan double limb requirement [99].

These low thresholds were then weaponised in the Government’s rhetoric. State governments and public health experts sceptical of the plan’s effectiveness were challenged with: ‘If not 70, when?’ [100,101]. The government slowly framed the policy choice as a crude dichotomy: either follow the plan and ‘live with COVID’ or face continuing lockdowns. The choice was presented as between the ‘heavy hand of governments and a ‘culture of mandates’ or a ‘culture of responsibility’ [102]. A more nuanced public health strategy including the use of masks, density limits and ventilation was erased from the agenda. Importantly the Morrison government’s slogan was ‘living with COVID’ not ‘living safely with COVID’.

The policy rhetoric thus shifted after the Plan’s release, especially from the Commonwealth and the Liberal New South Wales governments. The message now was that, with vaccination, all other restrictions could be removed. This was a risky strategy, dubbed ‘vaccine roulette’ [103], and was contrary to the ‘vaccine+’ strategy (supplement vaccines with a continuation of the other public health measures such as mask mandates) recommended by most public health experts. In retrospect, the release of the National Plan marked an important transition: public health measures began to be relaxed regardless of infection prevalence. It marked the end of the ‘control’, or ‘elimination’, era, and a shift to the ‘live with COVID’ era and the political acceptance of the death rate that entailed.

Although the published plan included the caveat that ‘The Plan is based on the current situation and is subject to change if required’, it did not change as circumstances did. Even at the most trivial level, for example, the 16+ age denominator for opening was not adjusted when children aged 12–15 became eligible for vaccination (announcement 27 August 2021, eligibility date 13 September) which meant that the Plan thresholds were no longer based on the population eligible for vaccination. More importantly, the SARS-CoV-2 virus mutates; each subsequent variant has different characteristics in terms of transmissibility and virulence, potentially making herd immunity through vaccination or infection not achievable [104]. This characteristic of the virus required the continuation of public health measures to mitigate the effect of virus spread.

Sriharan et al. [33] identified ‘adaptive competency’ as a key leadership competency because of the dynamic nature of pandemics. The National Plan contained a caveat about the possibility of changes in response to circumstances, but no changes were ever made, even when Omicron, a significantly more transmissible variant than Delta, became dominant. The lack of adaptive competency was demonstrated in this failure to adapt the Plan to a changing environment.

A key narrative during the pandemic, especially among those advocating a ‘live with COVID’ approach, was that COVID-19 was no worse than a bad flu season [105,106]. The Doherty Institute modelling, used by the Commonwealth Government in support of the 70 and 80 per cent 16+ coverage targets, was not aimed at eliminating infections, but rather ‘minimisation of moderate and severe health outcomes, defined as all identified cases leading to workforce absenteeism as well as that subset resulting in hospitalisation, intensive care requirement and death’ [95].

In the worst-case scenario in the Doherty modelling, relaxing certain restrictions at the 70 per cent 16+ threshold would be expected to lead to fewer than 20 deaths on any day in the three months after opening up, with an estimated 2710 deaths after six months. These daily and total worst-case estimates were quickly exceeded in 2021 but the Plan was not revised accordingly. There were more than six times the number of COVID-19 deaths in the first eight months after the Plan was unveiled than the 932 that had occurred in the 18 months prior to the Plan’s release. In the first four months of 2022, there were only 25 days when the death toll was under 20.

The effect of waning vaccine efficacy was even more significant. The Doherty Institute modelling, and the Government’s plans, were framed in terms of a fully vaccinated population, with ‘fully vaccinated’ defined as two vaccine doses. Against Delta, two doses of the vaccines available in Australia were very protective against infection, hospitalisation, and death. However, vaccine effectiveness waned [107,108] and so two doses no longer provided adequate protection against hospitalisation. In an Omicron environment, two doses more than 180 days prior provided 57 per cent protection compared to 90 per cent for a third dose [109].

In other words, the 70 and 80 percent two-dose vaccine thresholds no longer provided the level of protection modelled by Doherty. Rather than having a monotonically increasing level of vaccine protection in the country as increasing numbers of people were vaccinated, effective vaccine protection actually slipped backwards, thanks to waning and the low level of third-dose take-up.

On 10 February 2022 the relevant government immunisation advisory group (ATAGI) recommended that people 16 and over have a third dose, and if they had not had one within six months of their second dose, they be considered ‘overdue’—rather than not fully vaccinated, thus not triggering a breach of state public health measures which required ‘full vaccination’. By then, a majority of the public had recognised the changed reality, and supported defining ‘fully vaccinated’ as having a third dose [110]. It is unclear whether the advisory group advice was proffered truly independently, or whether its advice was influenced by signals from the Commonwealth government that it preferred an effective lower vaccination target rather than adopting a target based on a three-dose requirement. The retention of a two-dose definition weakened the case for reintroducing state public health measures and external border controls, based on the now prevailing lower effective vaccination rates than the National Plan thresholds.

The Commonwealth Government persisted with the outdated two-dose definition and the National Plan, its thresholds, and the definition of ‘fully vaccinated’, were not revised, even in the face of a rising death toll.

The Morrison government failed to provide effective national leadership during the pandemic. The failure to adapt the National Plan in the light of changing circumstances was driven by ideology and politics. The Government consistently attacked and undermined state public health measures and could not pivot from that stance in the light of the change in the environment and the reduction in effective vaccination protection.

However, Commonwealth government ministers should not be held solely accountable for all the failures of national leadership and the national plan, the bureaucratic processes also failed. Although different people can interpret evidence differently, and different decision makers may make different risk-risk trade-offs, the chief health officers, through their Principal Committee could have invested greater energy in consensus building, reaching agreement as far as possible on what the evidence was showing, and attempted to narrow bureaucratic disagreement to increase the chances of political agreement.

## 4. Managing External Borders

The Australian constitution vests control of external borders in the Commonwealth government. The early spread of the pandemic was mitigated by the closure of Australia’s external borders.

Border control, especially in the context of controlling maritime arrivals of asylum seekers, has been a key political issue in Australia for decades [111]. Externalising the pandemic threat, potentially reducing the need for internal controls [112], fitted neatly into the Morrison government’s playbook. Initially, Australians saw the SARS-CoV-2 virus as a ‘far-away threat’ [113] and were happy to keep it that way.

The World Health Organization does not support border closures as a response to a pandemic [114,115]. Border closures come with a human and economic cost [116] and so their use is controversial [117]. However, the evidence from both before [118,119], and in response to, the SARS-CoV-2 pandemic is that border closures reduce spread [120].

As recently as 2019, the Commonwealth Department of Health saw no place for international travel restrictions or mass quarantining of arrivals as part of a strategy of managing a pandemic [121] and had no plan to implement border restrictions [121]. Nevertheless, then Commonwealth Chief Medical Officer Dr Brendan Murphy recommended a border closure to China a few weeks after the virus was identified, and the Government announced the China border closure on 31 January 2020. This reduced the number of infected people arriving in Australia and is estimated to have delayed widespread transmission of the virus by about 4–6 weeks [122,123].

From mid-March 2020 the Government also required all arrivals—including returning Australians—to self-isolate and then later, to be quarantined for a two-week period, the estimated length of time thought necesary to minimise infection and transmission risk for arrivals [121]. Returning Australians had to pay for their stays in quarantine. Caps on international arrivals—matched to quarantine availability—forced airlines to cancel flights and limit passengers on flights which did arrive. Airlines increased prices to compensate for reduced load, making return unaffordable for many, and creating significant inequity in returns, with some able to enter on private jets and some left stranded [124], with complex advice from the Australian government about accessing support in their predicament [125]. This enforced separation between family members in Australia and those stranded overseas contributed to distress and anxiety on both sides [126,127]. The Department of Foreign Affairs and Trade was poorly prepared to deal with the consular issues associated with travel bans and was not able to meet the government’s expectations about providing appropriate assistance to the ‘stranded Aussies’ [128]. In response to a worsening outbreak in India, the government completely banned arrivals from that country, even of Australian citizens, an ethically challenging decision of dubious legality in terms of international law, but found to be legal under Australian law [129,130,131,132,133].

Although the Australian Constitution gives pre-eminence to the Commonwealth parliament over issues of quarantine, states can continue to make laws about quarantine provided they are not in conflict with Commonwealth laws [25]. The Commonwealth had dismantled its quarantine stations in the 1980s and so had no quarantine capacity itself, nor was it willing to build any. This meant that new arrivals were accommodated in state-managed ‘quarantine hotels’ [134]. These hotels could not control aerosol transmission of the virus, which, even at that early stage in the pandemic, was identified as a potential risk factor in transmission [135,136]. With the exception of the Howard Springs facility in the Northern Territory, the quarantine hotels were located in large population centres, which made leakage of infections more likely. Despite the obvious advantages of purpose-built quarantine facilities [18,137,138], the Commonwealth was slow to support states to build them.

Overall, the Morrison Government made the right call on the initial border closure, but its subsequent pandemic management was weak, particularly in terms of supporting quarantine and managing the return of Australians stranded overseas.

## 5. Protecting Residents of Residential Aged Care Facilities

Since 2011 the Commonwealth has had sole funding and regulatory responsibility for aged care [139] and could therefore be expected to have been proactive in managing providers’ preparation for, and response to, the pandemic. It was not.

Residential aged care outcomes were tragic: 7 per cent of all COVID-19 cases and 75 per cent of all deaths in the first year of the pandemic were of people living in residential aged care facilities [10]. The Morrison government attempted to play down the significance of the aged care death toll (and the death toll of older Australians in the community), qualifying announcements of deaths by referring to the approximately 40 per cent of the population [140] who ‘had underlying conditions’, or claiming that many of those in residential care who died were ‘palliative’ [141]. This approach represents an ageist failure to demonstrate empathy, a critical ‘people competency’ required in a pandemic [33].

The Morrison government’s aged care response was subject to excoriating criticism in a special Report from the Royal Commission into Aged Care Quality and Safety [142], and by independent academics [143]. Even late in the pandemic, the Government had no plan to manage the impact of the pandemic in aged care, or to provide advice to individual residential aged care facilities about how they might manage outbreaks [142,144]. Independent reviewers of the management of outbreaks also identified numerous weaknesses in the Commonwealth response [145,146,147]. The failure to plan shows a lack of the critical ‘task competency’ in the management of the aged care response.

The Morrison government’s management of COVID-19 in aged care is discussed in more detail elsewhere [148].

## 6. Personal Protective Equipment, Tests, and Vaccines

The fourth key responsibility for the Commonwealth Government was to approve, procure, and distribute personal protective equipment, tests, and vaccines. Again, this was an area where the Morrison government fell short, particularly in the case of vaccines. Its management of this key role showed it fell short on two key competencies: its poor vaccine procurement strategy demonstrating a failure of adaptive and task competencies, and its messaging also showing weak task competencies.

### 6.1. Personal Protective Equipment

The pandemic quickly disrupted every element of global supply chains for personal protective equipment (PPE), a disruption exacerbated by ‘just-in-time’ procurement practices and the fact that a significant proportion of all global PPE was sourced from China [149,150]. All countries scrambled simultaneously to source supplies, or belatedly to develop manufacturing capacity, and shortages of PPE were common across the world in the early stages of the pandemic [151].

One area of failure was in the management of the national medical stockpile, the source for the initial urgent supply of PPE. The national stockpile had been allowed to run down immediately preceding the pandemic: in the four years up to 2019, the stockpile had an average value of just over $100 million, only half the average value during the decade prior to that [152]. The Department of Health did not have good systems in place prior to the pandemic for emergency procurement, and, possibly as a result, inconsistent due-diligence checks of suppliers impeded procurement effectiveness [153]. Personal protective equipment provided by one supplier, Aspen Medical, was recalled by the Therapeutic Goods Administration because items were not sterile and did not provide protection from liquid splashes (TGA recall notification RC-2021-RN-00980-1).

Allocation of PPE was prioritised appropriately [154] but many providers in both primary care and hospitals felt exposed because of the lack of PPE [155,156]. Even in mid-2020, PPE was in short supply in Australia and less than half requests for supply from the national stockpile were being met [16].

### 6.2. Testing

The first stage of pandemic control is testing to identify who is infected, so they can isolate or be quarantined to prevent further infections, and to allow tracing of their contacts [157]. ‘TTIQ’—for Testing, Tracing, Isolation, and Quarantine—became a ubiquitous acronym during the pandemic. Tracing, isolation, and quarantine became state responsibilities, carried out with varying effectiveness, with the Commonwealth’s role in this area limited to its COVID-Safe app. The app was much hyped as ‘one of the critical tools we will use to help protect the health of the community’ [158] but proved to be useless [159], and decommissioned in 2022.

Tests for infection with SARS-CoV-2 were developed quickly, but testing kits were initially in short supply globally. This led to novel strategies, such as pooling of samples to make more efficient use of available resources [160,161,162]. Overall, Australia had a good testing rate throughout the pandemic [163,164].

The Morrison government negotiated a generous payment with private diagnostic providers for the standard polymerase chain reaction (PCR) test which included supervised collection of the sample. The alternative method, self-collection, was used in the United Kingdom for symptomatic people and approved by the United States Food and Drug Administration [165,166,167,168], but was not introduced in Australia. The negotiated payment to private pathology providers was about double what was paid to state-owned public providers and led to soaring profits for the private companies [169]. This pathology deal was not best practice: it did not include any discounts for an increasing volume of tests [170], nor any standards for timely provision of results.

A new generation of testing, Rapid Antigen Tests (RATs), became available in mid-2020. Unlike PCR tests, RATs provided results within minutes, and started to be rolled out internationally, including in the UK and in European countries, in mid-late 2020 [84,171]. When RATs were eventually approved for use in Australia in late 2021, people had to pay for them. This was an economically irrational decision since the benefits of testing flowed to other people; those tested would isolate and not infect others, while the cost fell on the tested.

A perfect storm emerged in late 2021 when the more transmissible Omicron variant took hold. Public health protections were being wound back in line with the National Plan, and more people began to travel and gather over the Christmas break. Infections sky-rocketed, and so did demand for tests. COVID-19 testing became the 2021 Christmas barbecue stopper. Inordinate delays occurred as people waited hours for PCR tests and days for results [172,173]. Laboratories were overwhelmed and some tests were not reported on at all. Governments responded by relaxing testing rules and promoting use of RATs. In the absence of a nationally managed supply arrangement, unscrupulous providers rationed RATS with price gouging, which attracted the regulator’s attention [174].

The testing debacle of the summer of 2021–22 was the result of a failure of strategy in the previous eighteen months. Although RATs are not perfect, their use in population screening and asymptomatic testing had become widespread internationally by late 2020 [175,176,177]. The Therapeutic Goods Administration (TGA), the unit of the Commonwealth Department of Health which is the independent regulator, was slow to approve any change in a testing regime that had proved so lucrative for private pathology companies. Representatives of pathology providers had consistently argued against testing strategies, such as self-PCR testing and use of RATs, which would cut into their revenue streams [178].

The failure of the vaccine testing strategy was sheeted home to the Government by the head of the TGA in September 2021, before the full scale of the problem was publicly apparent:

Therapeutic Goods Administration boss John Skerritt pushed back responsibility for Australia’s go-slow approach to at-home COVID rests on the federal government. “… we can’t formally make an approval decision until we get a signal from the government,” Professor Skerritt said. “It’s a decision for the government. Firstly, when they feel an appropriate time is to commit such tests. …They have to make a decision, you know, when is it less of a big deal to start missing some positive cases, because we know these rapid antigen tests are pretty good, but they’re not as good as the gold standard PCR test… Asked if the delay in the roll out of at-home tests until vaccination rates in Australia were higher was a deliberate strategy, Professor Skerritt said “correct” [179].

### 6.3. Vaccines

Everything that could go wrong with Australia’s vaccination program did go wrong. The highly politicised procurement strategy put Australia at risk if vaccines fell over, which they did. The vaccination rollout began months after rollouts in the United Kingdom and the United States, which meant that by mid-2021 Australia was among the worst performers internationally in terms of proportion of the population vaccinated; initial vaccination targets were not met; the choice of vaccination channels was driven by politics; logistics and vaccination prioritisation failed; the rollout of vaccinations to residential aged care and disability facilities was a slow-moving tragedy; and communications about approvals and changes in approvals created uncertainty and fuelled vaccine hesitancy.

However, this litany of failure did not stop Health Minister Hunt, who determined the procurement strategy, from describing the vaccination outcomes in early October 2021 as having ‘exceeded our expectations’, and ‘a huge national achievement’ [180]. The public’s assessment was not so positive. The percentage of the population rating the ‘Federal government’s response to the COVID-19 outbreak’ as quite good or very good dropped shortly after the vaccine rollout started, falling from the high 60 or low 70 per cent approval range to the high 30 or low 40 per cent range in August 2021 [181]. The average proportion of the population rating performance very high halved after March 2021, dropping from 25 per cent to 13 per cent.

The Government blamed everyone except itself for the failures, which was itself another mistake: acknowledging fault appears to be a better political strategy in crises than attempting to deflect blame [182,183].

#### 6.3.1. Vaccine Procurement Strategy

Vaccine procurement was, at least initially, a media bonanza for the Morrison government. Media releases flowed regularly from Health Minister Hunt. The Prime Minister and the Health Minister availed themselves of every opportunity to be filmed watching agreements being signed, production lines started, vaccines unloaded from planes, and test tubes examined. This politicisation of the vaccine rollout went as far as having the Liberal Party logo included on vaccine announcements [184]. However, positive stories disappeared in 2021 when the public wanted vaccines in their arms, not more announcements.

During much of 2020, vaccines were only a promise. It was not clear whether a vaccine against SARS-CoV-2 would be developed, and if so, when. A huge number of research groups entered the race [185], partnering with manufacturers, adopting novel approaches to speed up the vaccine-development process and bring the vaccines to market, where very large rewards lay in prospect [186,187]. In an environment of uncertainty, the optimum strategy is to ‘hedge one’s bets’, as economics Nobel Laureate Harry Markowitz showed in his classic work on modern portfolio theory [188]. That is what most countries did, spreading their investments and plans over multiple technologies, from multiple countries.

A presentation to the Morrison Government by consultancy group McKinsey in August 2020 confirmed this as the international standard procurement strategy [189]. The Government ignored this evidence, instead primarily investing in a two-vaccine strategy. Both its preferred vaccines had strong Australian links: one vaccine was under development by a research group based at the University of Queensland, and the other was developed at Oxford University with support from pharmaceutical giant AstraZeneca, which committed to partner with CSL to manufacture the vaccine in Australia. Vaccines based on newer mRNA technologies, Pfizer and Moderna, were also ordered, but in trivial quantities.

This ‘vaccine nationalism’ strategy [190] may have been developed because of fear of breakdowns in supply chains if all vaccines had to be procured internationally, but it quickly unravelled. The University of Queensland vaccine failed to clear an early hurdle when Phase 1 trial participants were recorded as becoming HIV positive [191]. The AstraZeneca vaccine failure did not come until later (see ‘Messaging’ section below). Moreover, CSL’s early manufacturing targets were not met, delaying the rollout.

Because of the initial narrow procurement strategy, no vaccine alternative was immediately available in sufficient quantities when things went awry. The Government eventually seized on the Pfizer vaccine as a solution to supply and hesitancy problems, which led to a mad scramble to purchase supplies from Pfizer directly, and from other countries which had near-expiry Pfizer supplies and were prepared to sell them to Australia.

Geneva-based international commentators provide this apt, diplomatically couched, summary:

Observing the vaccine strategies of other high-income, high-resource countries over the same time makes it clear that Australia could have invested its money into vaccine acquisition more wisely… The choice of entering into advance-purchase agreements with only three manufacturers, apart from the candidates backed by the COVAX facility, was risky [192].

#### 6.3.2. The ‘Strollout’

The result of the Government’s poor procurement decisions was that, in contrast to other countries which started vaccination programs in late 2020, the Australian vaccination program started late (in February 2021), was implemented slowly, and faltered. The experience in most countries was that the rate-limiting factor for vaccine coverage was on the demand side, the result of vaccine hesitancy. In contrast, in Australia the rate-limiting factor was on the supply side. There just was not enough vaccines to meet demand until September 2021. Australia ranked last in the world in terms of proportion of the population vaccinated in the middle of 2021, and did not start to climb the rankings significantly until the supply problems were overcome later that year [193].

The vaccine rollout strategy was influenced by the political and epidemiological environment of late 2020 and early 2021. State public health measures, and border quarantine, had largely protected Australia from the ravages of the pandemic. Virus spread was contained by the states’ COVID-zero strategy, which aimed to eliminate the virus [140]. However, the state strategies focused the limelight on state premiers and public health officials who made daily media appearances to announce the number of infections and any changes in lockdown restrictions. This left only vaccine procurement and rollout as areas where the Commonwealth could exercise leadership in the health response and win political credit for protecting the health of Australians, unfettered by state governments’ pre-eminent public health role. However, even here the Government’s response was found wanting. Support for the vaccination rollout, for instance, did not extend to income support, an area of Commonwealth responsibility. Workers could not access their sick leave to get vaccinations, and the Morrison government did not require employers to give workers time-off to get a jab, nor did it introduce other support schemes, except in the case of aged care staff. Other jurisdictions took a very different approach [194].

The Government’s view at the time was that the virus was essentially under control, which created an air of complacency about how quickly the vaccination program should proceed once vaccines became available. In March, the Prime Minister famously excused the slow start and early missed targets by describing the vaccination rollout as ‘not a race’ [195]. Health Minister Hunt accepted it was a race, but ‘a marathon, not a sprint’ [196], perhaps reflecting a better briefing that incorporated language used internationally [197,198] and by Australian academic commentators [199].

Any perceived threat of outbreaks was de-emphasised in planning. The political assumption at the time was that even with a February start to the vaccine rollout, most Australians would be vaccinated by late 2021. The Government could then call an early election for November or December 2021 at which it would coast to victory basking in the electorate’s gratitude for a successful vaccination program and a return to a life close to the pre-pandemic normal.

However, despite adopting a slow strategy for the rollout, the Federal Government still set ambitious goals. Prime Minister Scott Morrison initially said the goal was to vaccinate 4 million people by the end of March [200]. Actual vaccinations fell 3.4 million short of the target [201]. The slow start to the program should have allowed time for better planning and logistics management, but in the first few months of the rollout every stated target was missed, and a mere fraction of the expected vaccinations were delivered.

### 6.4. Staging

The vaccine rollout staging was announced by the Prime Minister on 7 January 2021 [200] and included five stages:Stage 1a: Quarantine and border workers; frontline health care workers; aged care and disability care staff; and aged care and disability care residents (estimated 678,000 people);Stage 1b: Elderly adults aged 80 years and over; elderly adults aged 70–79 years; other health care workers; Aboriginal and Torres Strait Islander people > 55; younger adults with an underlying medical condition, including those with a disability; critical and high-risk workers including defence, police, fire, emergency services and meat processing (estimated 6,139,000 people);Stage 2a: Adults aged 60–69 years; adults aged 50–59 years; Aboriginal and Torres Strait Islander people 18–54; other critical and high-risk workers (estimated 6,570,000 people);Stage 2b: Balance of adult population (4,643,000 people);Stage 3: <18 if recommended (5,670,000 people).

The language in the Prime Minister’s media conference suggested the order was a sequence; a visual explainer showed that Stage 1a would essentially be completed before stage 2a commenced, so that those most at risk would be vaccinated first. However, the staging plan was ignored from the beginning. The Prime Minister, who was not eligible to be vaccinated at the time, was the second person vaccinated in Australia—perhaps to demonstrate trust in the vaccine, or perhaps as part of his political strategy to emphasise the Commonwealth’s responsibility in this area. In many other countries, political leaders (and royalty) were vaccinated in line with their assigned priority [202].

People in residential aged care and in disability accommodation, and staff in those facilities, were assigned to Stage 1a of the rollout plan, but later stages were initiated before a significant proportion of those in the early stages had been vaccinated, with tragic consequences. Despite the recognised high-risk status of those living and working in aged care and disability accommodation [203], poor oversight by the regulator, which failed to sound the alarm about low vaccination rates, combined with the Morrison government’s lack of concern about the low staff vaccination rate [204,205] and its victim-blaming approach—attributing low vaccination rates to resident and family choice [206]—exacerbated the problem.

Midway through the vaccination program, blame was shifted onto the Commonwealth Department of Health for the failures of the rollout. Administration of the program was militarised as part of an attempt to deflect political attention: in April 2021 a navy commander was appointed to direct the rollout [207]. An army general took over in June [208]. The Prime Minister later stated that he should have militarised earlier [209].

The polling data cited earlier show that the public still held the Government accountable. Not for nothing did ‘strollout’, a typically Australian and laconic critique of the Government’s performance, became the Australian 2021 word of the year.

### 6.5. Vaccination Distribution

A low prevalence of infection, coupled with the wish to claim credit, led the Commonwealth to eschew vaccine distribution by state governments. Instead, it prioritised distribution through channels it funded directly: vaccines were to be primarily distributed through general practices and pharmacies.

This approach unravelled when supply shortages meant that commitments to general practices to deliver a specific quantity of vaccines were not met, causing anger and frustration over cancelled appointments, and further undermining confidence in the availability of vaccines.

Failure to supplement the boutique general practice strategy with high-profile state-run mass vaccination centres as part of the distribution strategy—an approach adopted by other countries [210], and supported by modelling [211]—also slowed public uptake.

As the rollout went on, there was regular public reporting of the number and proportion of the eligible population vaccinated according to age classifications. However, there was very little reporting of progress in vaccinating people in the designated priority groups, such as aged care residents and workers, and people with a disability. As a result, low vaccination rates in these sectors were not identified early enough to take corrective action. Data were not routinely published by location of those vaccinated either, so gaps in vaccination coverage for low-income populations were also missed, and nothing was done to increase uptake in those groups. This exacerbated the inequitable adverse socio-economic impact of the pandemic.

The pandemic exacerbated the adverse impacts of pre-existing socio-economic drivers of ill-health. People in precarious employment or in jobs that required frequent human contact, were more exposed to the virus and more likely to be infected. A well-designed vaccination programme should not have shown a socioeconomic gradient or a differential take up of vaccinations in culturally or linguistically diverse communities. Specific strategies to ensure equitable take up across all communities were required to ensure this [212,213], and many were proposed for implementation in Australia [214,215,216,217].

Despite the forewarning of a potential socio-economic gradient, the government’s response was demonstrably ineffective: two-dose vaccine take up in the most-disadvantaged quintile was about two thirds that in more advantaged neighbourhoods [218], and, indeed, socioeconomic effects were seen across every dimension of disadvantage [219].

#### 6.5.1. Messaging

The communications about vaccine supply arrangements, and whether or not the public would have a free choice of vaccines, was poor from the start. The mainstay for the vaccination rollout was planned to be AstraZeneca but Health Minister Hunt caused some confusion when he initially suggested people unsure about AstraZeneca might wait until a different vaccine was available. He subsequently retracted those comments [220].

This ‘choice approach’ was consistent with the view that there was no urgency for the rollout. A strategy that allowed people choice was unusual in the circumstances, especially given supply issues associated with the Pfizer vaccine. Public preference for Pfizer became stronger as problems with the AstraZeneca vaccine became more apparent.

Reports of complications and deaths following the use of the AstraZeneca vaccine began to emerge in Europe in early 2021 [221], and while the other vaccines in use also reported adverse events, they did not attract the same media attention [222]. In Europe the risk-benefit trade-off was clear: given the prevalence of the virus, vaccination with AstraZeneca was lower-risk than infection [223]. The risk-benefit trade-off in Australia was more nuanced: governments and citizens needed to weigh the low contemporary community prevalence of SARS-CoV-2 against the risks, including the risks of new outbreaks and the constrained availability of alternatives to AstraZeneca.

Deaths associated with AstraZeneca were rare and primarily associated with a very rare genetic condition [224], but were nevertheless widely reported [225,226,227,228], contributing to vaccine hesitancy. The regulator [229], supported by the scientific community [230], attempted to reduce concerns, stressing the very small risk of death from vaccination. However, these *logos-*based arguments were poorly designed framed in a ‘public ignorance’ narrative [231]. They were rarely targeted at the specific subpopulations with low vaccine take-up. They did not draw on the science of risk perception and risk communication [232,233,234], nor did they address potential issues of trust, or—at the extreme—conspiracy theories [235].

In response to the accumulating evidence about adverse events, the relevant government advisory group, the Australian Technical Advisory Group on Immunisation (ATAGI), initially issued reassuring statements (25 March 2021, 2 April 2021), but shifted its position to a recommendation that Pfizer was ‘preferred’ for people aged under 50 (ATAGI statement of 8 April 2021). ATAGI’s cautious approach was informed both by the immediate risk-benefit ratio (including low contemporary prevalence of SARS-CoV-2) and the fact that the recommended alternative vaccine, Pfizer, was in short supply because of the procurement strategy failures.

As reports of adverse events accumulated, including seven cases in people aged 50–59 in the second week of June, ATAGI revised its advice on 17 June to preference Pfizer for people under 60. Within thirty minutes of ATAGI’s statement, Health Minister Hunt publicly announced it—but without any additional communications plan, and in apparent disregard of the fact that Pfizer was still in short supply [236]. Not surprisingly, public acceptance of AstraZeneca tanked, exacerbating supply constraints and fuelling vaccine hesitancy [98].

The 17 June ATAGI statement also included advice that

COVID-19 Vaccine AstraZeneca can be used in adults aged under 60 years for whom Comirnaty (Pfizer) is not available, the benefits are likely to outweigh the risks for that individual and the person has made an informed decision based on an understanding of the risks and benefits [237].

The Prime Minister subsequently used this nuanced ATAGI advice in a rambling and confusing late-night announcement [238] that encouraged younger people to be vaccinated with AstraZeneca, with their GP and themselves taking on the risk of adverse events. The Prime Minister’s statement was then very publicly opposed by then Queensland Chief Health Officer, Jeannette Young [239], creating further controversy. Risk messaging should be conveyed by trusted people [240,241], but trust in the Prime Minister had peaked in late 2020, and by mid-2021 was on a trajectory of decline in net trust [242].

Changing advice as additional information becomes available is the right call. However, the overall messaging in the first six months of the vaccination program was uncoordinated, sometimes confusing, and poorly designed. The absence of experts in risk communication hindered efforts to overcome developing vaccine hesitancy. The reasons for changed advice were poorly explained, with the result that trust in AstraZeneca plummeted, even among those aged 60 and above, just as infections were beginning to increase again with the Delta variant.

#### 6.5.2. Responsibility

The slow start and slow rollout of the vaccination program was costly. A better program could have saved lives, reduced the burden on health systems and staff [243], and averted an estimated $31 billion in damage to the economy [193]. The human costs of the bungled procurement and rollout was significant. With an alternate procurement strategy, the mid-2021 lockdowns could perhaps have been avoided, with weeks and perhaps months of fewer restrictions, fewer impacts on children, and fewer mental health issues.

Australia’s vaccination program has attracted a thesaurus of negative adjectives. The key issue now is understanding what caused these failures.

The Government obviously wanted to be a cleanskin: to not have to bear any of the responsibility. The creeping militarisation of administration of vaccine rollout in mid-2020 sent the signal that the public service was to blame, and could not manage the implementation. It also meant that tough questions could be deflected to a khaki spokesman.

The attribution of responsibility for the failure of the rollout to the Department of Health has also been advanced by independent commentators, who highlight the fact that successive governments, following the neoliberal playbook, had stripped out the policy and administrative capacity of the public service [190,244,245,246,247,248,249], an argument advanced to account for failure of the English COVID response too [250]. Others have suggested the failures were a consequence of the clientism approach [251] favoured by the Morrison government across a range of portfolios, including Health [252], and its tendency ‘to follow its ideological preferences for outsourcing to commercial entities’ [190]. However, there is a chicken and egg problem to disentangle here. Certainly the Government was quick to hire consultants to enable ‘overall program implementation monitoring’ and as ‘program delivery partners’ [253]. However, it is unclear whether consultants were hired because the Government preferred to heed advice from consultants (recall Minister Hunt had previously worked for McKinsey), or because the Department was incapable of providing timely advice.

The health quality improvement literature suggests that there is rarely a single cause for adverse events—it is usually a combination of contributing factors that leads to poor outcomes. So too with the vaccine program. The cascade of failure started with the policy design. As Yen, Liu, Won and Testriono [90] point out, ‘without appropriate policy designs, even the most capable state can fail in the face of a crisis’.

Advised by an expensive kleptocracy of consultants, the vaccine procurement strategy was clearly a political one, and consistent with the general Morrison government approach that ministers do policy and departments implement [254,255]. The politically-driven poor procurement decision was the critical distal cause. It led to the slow rollout because of the failure to procure sufficient vaccines, and it contributed to increased infections (and deaths) when the new, more transmissible and more virulent variants emerged and wreaked havoc in an only partially vaccinated population in late 2021.

Another strategic mistake was to emphasise a privatised Commonwealth-distribution strategy through GPs and pharmacists rather than embrace state distribution through mass vaccination hubs. Again, this appears to have been a politically-driven, credit-claiming choice that created logistical challenges.

The vast majority of Department of Health staff are based in Canberra, with only a few in state offices. Given that the Department essentially has no feet on the ground, and so no national implementation capacity, everything it does is done through private providers or states. The government’s vaccine distribution strategy initially relied entirely on the former channel. The private, small-provider strategy was a very risky one. The subsequent failure of implementation was almost inevitable, given the strategic choices.

The testing debacle was another clearly driven by political decisions, potentially the result of lobbying by pathology company rent-seekers [256]. The head of the Therapeutic Goods Administration called out the political nature of the process when he admitted that having to wait for government signals was the reason for slow decision-making [179].

The public health policies pursued by the Commonwealth were often in competition with those pursued by the states, and state public health measures were undermined as a result. The different Commonwealth strategy was, again, politically and ideologically driven, reflecting different priorities about the importance of business and economic interests compared to public health concerns. These differences could perhaps have been mitigated if the Australian Health Protection Principal Committee (the meeting of the chief health officers) had been more effective as a dispute settling mechanism, and a forum where the quite different cultures of state and Commonwealth approaches to service provision and funding could have been understood better by the other party, and joint strategies developed.

Finally, the communication failures were mainly of politicians’ making, although conflicting messaging from state chief health officers probably contributed to confusion and uncertainty.

It is too easy to see problems on the ground and assume that the failures are all administrative. It is important to consider the strategic choices that increased the likelihood of implementation failure.

Nonetheless, there were obvious bureaucratic failures. The aged care regulator was trenchantly criticised by the Aged Care Royal Commission for failing to take sufficient action when outbreaks occurred, a continuing failure that is still in evidence [257] at the time of writing (mid 2022). Freedom of Information requests to shine light on the nature of Departmental advice have produced documents so heavily redacted it is not possible to disentangle what advice was given about strategic choices, and whether that advice was heeded [258].

However, even with such limited information about what was happening inside the bureaucracy, there are clear areas where bureaucratic processes hindered the pandemic response. Why could the key groups of officials and experts (the Australian Technical Advisory Group on Immunisation, and the Australian Health Protection Principal Committee), as experts, not provide advice on the many matters where jurisdictions went their own ways? Why did the experts in procurement in the Department of Health not do a better job on personal protective equipment, vaccines, and tests? The need for urgency cannot provide the full answer to this.

## 7. Where to from Here?

Many countries found it difficult to adapt and respond to the SARS-CoV-2 pandemic, and all countries are looking to learn how to handle shocks in the future [259,260].

Australia, got a lot right, including the strong public health measures introduced by states and the Commonwealth’s external border closures, resulting in significant numbers of deaths averted in 2020 and 2021. However, the Commonwealth government’s performance overall was not good. The Morrison government failed in three of its four key pandemic responsibilities. National leadership was poor, impeding and undermining the public health response; the aged care response was a tragedy; and the Commonwealth failed to ensure adequate supply and distribution of personal protective equipment, tests and vaccines. The only bright spot was border control, and even here the plight of stranded Australians was not handled well.

The key immediate problems underlying this sorry saga are politicisation and politically driven decision-making, with the consequential failures in implementation, and issues related to Commonwealth–state coordination. 

The failures of implementation may not have been inevitable or as egregious if the initial strategic choices had not been so political. The politicisation of the Commonwealth response affected every element of the Morrison Government’s approach. The Government pledged to under-promise and over-deliver [261], but did the opposite. Its relentlessly optimistic political messaging made it harder for the Government to admit its mistakes, learn from them, and reset its agenda.

One way of limiting politicised decision-making may be to make it more transparent. Although the mantra of ‘following the expert advice’ was regularly invoked during the height of the pandemic, it is patently clear that it was not always followed. Evidence, though, can and should be contested [262], and academic research has to be processed or ‘transformed’ to be incorporated into policy [263,264]. ‘Following the evidence’ involves making judgements, especially in the early stages of a pandemic when evidence is developing rapidly, about what studies to accept or weigh highly and what not to. Decisions about public health measures involve managing risks of adverse consequences of the measures, which fall unevenly, and the benefits, which also fall unevenly, and all of this when the evidence may not be clear cut.

Recent changes in Victoria may, with appropriate adaptation [265], provide a model for reform at the federal level to make value assessments of evidence and its implications clearer. The changes to the Victorian *Public Health and Wellbeing Act* provide that decision-making in public health emergencies sits with the Premier, thus ensuring public accountability. The Chief Health Officer, who previously held these powers, must be consulted and provide written advice to be tabled in Parliament. The reasons for the Premier’s decisions must also be tabled (Section 165AG).

The second underlying cause of the problems during the pandemic related to the need to coordinate decisions of the Commonwealth and the states. States share boundaries and New South Wales’ decisions to open up allowed the virus to spread to Victoria.

But coordination is one of the most fraudulent words of politics and administration. It dresses neutrally to disguise what nakedly is pure political form. Coordination is a political process by which the coordinated are made to change their value positions, their policy conceptions and their behaviour to conform to the conceptions and expectations of the coordinator [266].

The Commonwealth clearly resented the fact that states were not compliant politicizat. The failure of the National Cabinet meant that Australia effectively had ‘low state capacity’—that is, it could not coherently or effectively mobilise the instruments of state capacity to manage the pandemic [90]. That Australia performed so well, at least up until the release of the National Plan, is a tribute to the strengths of the states.

The politicization problem and the Commonwealth-state issues may have a common solution: an independent Centre for Prevention and Disease Control (CPDC), which could be created through a new intergovernmental agreement, with supporting legislation.

Increasingly advocated by independent commentators [267,268,269,270] and supported by Labor at the election [271], a CPDC is not a panacea. In America, for instance, the actions of the United States’ Centers for Disease Control were hindered by the COVID-denialism of the Trump administration [272], and various attempts to sideline it [273,274,275]. However, as Table 2 illustrates, a well-designed CPDC could address critical problems identified in the management of the pandemic in Australia in 2020 and 2021, and could be used to build state capacity to prepare for the next pandemic.

A new CPDC would have to operate in a crowded and contested space. It should not be framed as an ‘infectious disease’ agency, but as part of the national capacity to address or prevent both infectious and chronic diseases, with a broad remit. Some of the communication weaknesses seen in the response to the pandemic might have been averted if a national agency existed which advocated and followed the US CDC’s six Crisis Emergency Risk Communication principles: Be first, Be right, Be credible, Express empathy, Promote action, Show respect [276], or indeed if the existing relevant bodies (such as the Australian Technical Advisory Group on Immunisation) did so.

The immediate problems discussed above are easily addressed. However, the critical issue is that Australia was not well prepared for the SARS-CoV-2 pandemic, which is ‘to a large extent a neoliberal pandemic’ [277], the first Australia has faced since the downsizing of public sector capacity. Others have also linked the pandemic to neo-liberalism [278,279,280,281]. However, neoliberalism is a contested concept, plagued by conceptual confusion [282,283]. Navarro has clarified three core elements of the neoliberal ideology:

(1) the state (or what is wrongly referred to as government) is part of the problem rather than the solution and needs to be reduced; (2) labor and financial markets need to be deregulated in order to liberate what is defined as ‘the enormous creativity of the markets’; and, (3) commerce and investments need to be stimulated by eliminating borders and barriers to allow the full mobility of labor, capital, goods and services [284].

The first and last of these had a specific impact on the COVID pandemic.

The Morrison government’s response to COVID-19 was taken directly from this neo-liberal playbook. The economy was to be prioritised in an imaginary tradeoff between health and economics. Private sector consultants’ advice was prioritised over public service advice. The ‘productive’ members of the economy were to be prioritised over those with ‘underlying conditions’.

The Commonwealth government had a choice of its narrative: would it emphasise the benefits of collective action or emphasise individual freedom and the economy? It chose the latter, weakening support for any continued collective action and contributing to the excess deaths from the pandemic in 2022.

Addressing the underlying neo-liberal cause of the Morrison government’s failed response to the pandemic will not be easy but should be attempted. The neo-liberal paradigm, currently so dominant globally, must be challenged as its weaknesses are revealed [250,278,285]. The new Albanese government has the opportunity to do this and reset the agenda. It must value and rebuild the public service, saving money by reducing wasteful spending on consultants. It can use the power of rhetoric too, talking up social solidarity in public responses to public problems and being unafraid to talk about social and economic determinants of health and taking action on addressing them. The neo-liberal verities of ‘there is no such thing as society’ [286], ‘government is the problem’ [287], and ‘steering not rowing’ [288] need to be challenged and replaced by and emphasis on the benefits of public provision and that government is the local nurse, teacher, bus driver, or police officer. Rebuilding confidence and trust in government is a long term strategy, but a necessary one to prepare for future health challenges.

## 8. Conclusions

This paper presents a comprehensive assessment of Australia’s national government’s response to the SARS-CoV-2 pandemic. It has three main limitations. Firstly, it relies on publicly available information. The author does not have access to internal discussions or documents about who recommended what, or why certain decisions were made. Secondly, it was written in the first half of 2022, before the pandemic has run its course. New information about the internal operations of the Morrison government, and how and why decisions were made, is becoming available almost daily. New analyses of the impact of the pandemic in Australia are also being published. These have not been considered. Thirdly, the author was a participant in the public debate about the pandemic, supporting a ‘COVID-zero’ approach (Duckett, Mackey and Chen 2020). The potential for bias generated by that involvement is somewhat mitigated by the extensive referencing of other sources adopted in this paper. Earlier drafts of this paper were also shared with others and presented at university seminars to get feedback on the diagnoses and conclusions, and to identify areas where evidence was weak or conclusions unfair. The refereeing process for this publication also assisted to ensure a balanced evaluation.

Australia’s decentralised public health system mostly served the country well. Despite the manifest failure in national political leadership and poor strategic choices, the overall outcomes were good. The states stepped up and provided leadership and made the tough decisions that controlled the virus and delivered the outcomes that led to international praise. However, states also failed in some aspects of their management of the pandemic as mentioned earlier.

The Morrison government’s record in management of the pandemic was very poor as demonstrated by the absence of the critical crisis leadership competencies—task, adaptive, and people competencies [33] with significant failure seen across three of its four roles. Unfortunately, the Morrison government’s failures did not end with its defeat at the polls. Its relentless undermining of state public health measures and dichotomising the possible pandemic responses into lockdowns or not, has weakened states’ social licence to use even relatively unobtrusive measures such as mask mandates to respond to the ongoing pandemic, thus contributing to the continuing deaths and hospitalisations.

Lessons must be learnt, and new systems developed to ensure that the weaknesses exposed during the pandemic provide a basis for better planning and better pandemic responses in the future.

## Figures and Tables

**Figure 1 ijerph-19-10400-f001:**
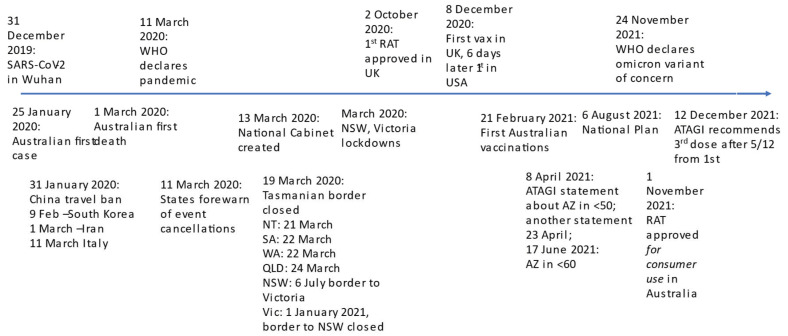
Key COVID-19 dates. Source: Author’s analysis.

**Table 1 ijerph-19-10400-t001:** COVID-19 total cases and deaths as of 30 April 2022.

	Cases		Deaths	
	Number (Million)	Per Million Population	Number	Per Million Population
Australia	5.95	230,847	7250	281
Canada	3.77	98,921	39,312	1033
New Zealand	0.937	182,724	707	138
United Kingdom	22.11	324,199	175,011	2567
United States	81.35	244,346	993,712	2985

Data source: https://ourworldindata.org/.

**Table 2 ijerph-19-10400-t002:** The role of a Centre for Prevention and Disease Control in addressing problems seen in national management of the SARS-CoV-2 pandemic.

Problem	Contribution of CPDC to Solution
Politicisation of decision-making	Independent board of CPDC with board members appointed by both the Commonwealth and state governments (Independent Hospital Pricing Authority might provide a model) Charter of independence (Reserve Bank might provide a model)
Lack of transparency of decision-making	Requirement to publish advice to the ministers Commonwealth and state ministers to respond within designated timeframes
Equity issues not managed well in the pandemic	Give CPDC explicit responsibility for addressing social and economic determinants of health status and outcomes, and leading Australia’s response in these areas (not only during pandemics)

Source: Author’s analysis.

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
