# Peer review of "Public Health Management of the COVID-19 Pandemic in Australia: The Role of the Morrison Government"

_ijerph, 2022, doi:10.3390/ijerph191610400_

Round 1

Reviewer 1 Report

This is an impressive and detailed review of the management of the COVID 19 pandemic in Australia with a focus on the role of the national government, drawing together an impressive array of research and commentary. It is not, however, as balanced as it might be.

For a more balanced presentation, I suggest first some more information be presented early in the introductory section on ‘A new virus – four roles of the national government’, describing briefly the constitutional roles of the Commonwealth and the States as they have evolved. This would identify ‘quarantine’ as a Commonwealth power ever since federation and pharmaceutical and hospital benefits and medical and dental services as a Commonwealth power since 1946. In more recent decades, the Commonwealth has had primary responsibility for public funding of the health system and its broad policy design while the States have responsibility for the delivery of public hospital and most public health services; the Commonwealth has also gained almost exclusive responsibility for public funding and regulation of aged care.

Other suggestions to ensure a better balance are included in the comments below on the major sections of the article.

Leadership

The balance might be improved if the section on ‘The National Cabinet’ briefly described the machinery under both the National Cabinet and its more accurately named predecessor, the Council of Australian Governments’, including AHMC and AHMAC and their committees including the AHPPC and ATAGI. This machinery highlights how the Commonwealth and the States and Territories are meant to cooperate and collaborate. In many respects, it was the failure of this machinery, not just the mistakes of the Commonwealth ministers (the PM and Health Minister in particular), or some of the Premiers, that underpins the weaknesses of the Australian response (this is only touched upon towards the end of the article on p17).

The timeline set out in Figure 1 is also central to the assessment of the Australian response and who performed well or poorly, and when.

The material on pp4-5 understates the level of cooperation in the early period (until later in 2020). The National Cabinet arguably operated quite well at that time according to all the participating First Ministers. The PM also was appearing regularly with the CMO, as Premiers and Chief Ministers were appearing with their CHOs, emphasising the extent to which political leaders at both levels were taking seriously expert health advice, and that the health advice was being coordinated through the AHPPC. There was little debate at that time about whether public health measures were inconsistent with economic advice.

This consensus began to unravel later in 2020 and, as the article shows, the claim of a trade-off between public health and the economy was promoted by Commonwealth ministers before any such trade-off genuinely arose (as vaccination rates increased and the costs compared to the benefits of border closures etc. became less clear). The unravelling may also have reflected the beginnings of failures of the AHPPC to ensure a common officials’ presentation of expert health advice (after appropriate deliberation of the evidence).

The assessment of the National Plan (pp6-7) is well argued, but the failure in February 2022 to adjust the definition of ‘full vaccination’ was surely the fault of ATAGI not just the Commonwealth Government which ‘persisted with the out-dated two-dose definition’.

The assessment of the management of the external boundaries is also well argued and the conclusion drawn (p9) seems fair, distinguishing between the Commonwealth’s early and later efforts. It may well be that the Commonwealth’s predisposition not to rely upon government provision of services contributed to the lack of adequate quarantine facilities.

Aged care

Given the reference to Lacey and Prosser 2022, the article only briefly discusses the key issue of protecting aged care residents. The failures here certainly reflect inadequate support and regulation by the Commonwealth. It is hard not to conclude, however, that much of the problem lay with officials in the Department of Health, not just ministers (and the political leadership).

PPE, tests and vaccines

While most of the assessment of Australia’s performance with regard to testing and the vaccination program is well supported in the article, with Commonwealth weaknesses contributing most to the failures, in some places blame is too hastily directed at the political leaders (the reference on p13 to an intended election in late 2021 is surely just speculative, and the PM’s own early vaccination (p14) might have been to highlight his advocacy and how easy vaccination is, even if he was not yet eligible); and it is not sufficiently directed to Health officials including at the State level. The ATAGI advice about A-Z and Pfizer was, as mentioned on p15, based on cost-benefit risk assessments when the prevalence of the virus was low in Australia: that was clearly an error as the likely spread of the virus was surely well known (Ergas and Pincus 2021). This error was exacerbated by the public statement from the Queensland CHO highlighting the risk of death among younger people from A-Z vaccination. While Minister Hunt’s announcement in June 2021 did not help, most of the blame for public hesitancy of A-Z, and limiting its use, must surely lie with officials.

The discussion of ‘Responsibility’ for the vaccination roll-out (pp16-17) is mostly supported by the evidence presented though the extent to which the failures were ‘politically driven’ is perhaps exaggerated. There were failures amongst the bureaucrats and public health advisers and some of the political decisions were not entirely mistaken (using GPs and pharmacists did have advantages but should have been complemented by major government-run vaccination centres). The different cultures of the Commonwealth and State officials may also have played a part (one working mostly through private providers and the other more familiar with direct service delivery), a problem that should have been worked out through AHMAC as well as the AHPPC.

Where to from here?

To say that ‘Australia, led by the state Premiers, got a lot right’ (p17) is too strong, as is the statement on p19 that ‘The States stepped up and provided leadership and made the tough decisions that controlled the virus and delivered the outcomes that led to international praise’. The Commonwealth got some things right, and the State Premiers got some things wrong; arguably the Commonwealth’s errors were more serious in the end, though Australia’s overall performance in terms of deaths per million is still lower than that of most OECD countries.

The proposal for an Independent Centre for Prevention and Disease Control nonetheless makes considerable sense, to give more emphasis to expert advice and to drive better collaboration amongst jurisdictions’ health advisers.  Mention might be made of some of the strategies used by the US CDCP which an Australian CPDC should follow, and might have helped the AHPPC and ATAGI if applied during the pandemic. These include six Crisis Emergency Risk Communication principles: Be first, Be right, Be credible, Express empathy, Promote action, Show respect (CDCP 2018). I doubt, however, that the location of a CPDC matters much (p19).

The emphasis toward the end of this section and in the conclusion on ‘neo-liberalism’ as the underlying cause is overstated and over-simplistic. The term is not defined (Shergold and Podger 2021) in any case. Certainly ‘politicisation’ was a factor along with ‘externalisation’ (Halligan 2020) emphasising the role of non-government providers; both have contributed to loss of public sector capability and both almost certainly coloured Commonwealth ministers’ attitudes towards firm government action and direct service delivery by government.

Minor issues

The layout of the article needs attention. It is not clear when the Abstract ends, and major and minor headings need distinguishing.

The reference to Jobkeeper (p5) could be better presented. Some acknowledgement of its importance not only to ensuring income protection but also to maintaining employment is needed. The weakness however might also be strengthened by reference to the fact that Jobkeeper provided excessive support to businesses not adversely affected by the pandemic.

REFERENCES

Center for Disease Control and Prevention, 2018. Crisis and Emergency Risk Communication Introduction, Washington.

Ergas, Henry and Jonathan Pincus, 2021. ‘What to do about shifting AZ advice’, The Australian, 27 July 2021

Halligan, John, 2020. Reforming Public Management and Governance: Impact and Lessons from Anglophone Countries, Edward Elgar, Cheltenham

Shergold, Peter, and Andrew Podger, 2021. ‘Neoliberalism? That’s not how the practitioners view public sector reform’, in Podger et al (eds), 2021, Politics, Policy and Public Administration in Theory and Practice, ANU Press, Canberra

Author Response

Thanks for the very useful comments

Reviewer 2 Report

The aim of this paper is to analyse how the Morrison government managed its health sector-related COVID-responsibilities. The article uses case studies to fully demonstrate the Morrison government's lack of capacity in COVID-19 governance from the perspective of the four functions of government. It further analyses the reasons for the lack of capacity and makes recommendations for improvement.

Major comments:

1.       The article does not indicate the main marginal contributions. There is a need to summarise existing studies and present the marginal contributions of the article.

2.       Please briefly describe the method this paper used.

3.       In the section “A new virus—four roles for the national government”, there are four health-related responsibilities. Is there a good reason why these four roles were chosen? What about other responsibilities? For example, focusing on health inequalities for vulnerable groups, protecting workers in the workplace to avoid risk of exposure to COVID-19.

4.       In the fourth role, 'Personal protective equipment, tests, and vaccines', the argument does not indicate which competencies the government lacks in this role. It would have been clearer if the section had begun or ended with a summary of what competencies the government lacks, corresponding to the case studies.

5.        The conclusion should include limitations.

Some critical formal issued should be revised as well:

1.       The keywords are missing. (After abstract)

2.       There are some sentences that are incomplete or have been subdivided in the middle, please check. For example, line 128, line177, line 894-line895.

3.       There is no title for the first section

Author Response

Thanks
